# The Effects of Ozone on Atlantic Salmon Post-Smolt in Brackish Water—Establishing Welfare Indicators and Thresholds

**DOI:** 10.3390/ijms21145109

**Published:** 2020-07-20

**Authors:** Kevin T. Stiller, Jelena Kolarevic, Carlo C. Lazado, Jascha Gerwins, Christopher Good, Steven T. Summerfelt, Vasco C. Mota, Åsa M. O. Espmark

**Affiliations:** 1Nofima AS, NO 9291 Tromsø, Norway; jelena.kolarevic@nofima.no (J.K.); carlo.lazado@nofima.no (C.C.L.); Jascha.Gerwins@Nofima.no (J.G.); vasco.mota@nofima.no (V.C.M.); asa.espmark@nofima.no (Å.M.O.E.); 2The Conservation Fund’s Freshwater Institute, Shepherdstown, WV 25443, USA; cgood@conservationfund.org (C.G.); steve@superiorfresh.com (S.T.S.)

**Keywords:** aquaculture, ozone, Atlantic salmon, brackish water, gill health

## Abstract

Ozone is a strong oxidant, and its use in aquaculture has been shown to improve water quality and fish health. At present, it is predominantly used in freshwater systems due to the high risk of toxic residual oxidant exposure in brackish water and seawater. Here, we report the effects of ozone on Atlantic salmon (*Salmo salar*) post-smolts (~100 g), in a brackish water (12 ppt) flow-through system. Salmon were exposed to oxidation reduction potential concentrations of 250 mV (control), 280 mV (low), 350 mV (medium), 425 mV (high) and 500 mV (very high). The physiological impacts of ozone were characterized by blood biochemical profiling, histopathologic examination and gene expression analysis in skin and gills. Fish exposed to 425 mV and higher showed ≥33% cumulative mortality in less than 10 days. No significant mortalities were recorded in the remaining groups. The skin surface quality and the thickness of the dermal and epidermal layers were not significantly affected by the treatments. On the other hand, gill histopathology showed the adverse effects of increasing ozone doses and the changes were more pronounced in the group exposed to 350 mV and higher. Cases of gill damages such as necrosis, lamellar fusion and hypertrophy were prevalent in the high and very high groups. Expression profiling of key biomarkers for mucosal health supported the histology results, showing that gills were significantly more affected by higher ozone doses compared to the skin. Increasing ozone doses triggered anti-oxidative stress and inflammatory responses in the gills, where transcript levels of *glutathione reductase*, *copper/zinc superoxide dismutase*, *interleukin 1β* and *interleukin* were significantly elevated. *Heat shock protein 70* was significantly upregulated in the skin of fish exposed to 350 mV and higher. *Bcl-2 associated x protein* was the only gene marker that was significantly upregulated by increasing ozone doses in both mucosal tissues. In conclusion, the study revealed that short-term exposure to ozone at concentrations higher than 350 mV in salmon in brackish water resulted in significant health and welfare consequences, including mortality and gill damages. The results of the study will be valuable in developing water treatment protocols for salmon farming.

## 1. Introduction

An increasing number of Atlantic salmon (*Salmo salar*) producers are raising fish up to 1 kg (post-smolts) or even to complete market size in recirculating aquaculture systems (RAS) [1,2]. It has been recently shown that Atlantic salmon post-smolt in RAS have better growth and welfare in brackish water (12 ppt) compared to seawater (32 ppt) [3]. RAS suppliers and users are searching for safe, cost-efficient and reliable ways to maintain optimal water quality in these systems [4,5,6]).

Ozone (O_3_) is a clear blue colored gas that is formed when an oxygen molecule (O_2_) bonds with a third oxygen atom (O). It is highly unstable, making it a potent oxidizing agent with high germicidal effectiveness and is thus ideal for use in water treatment. Ozone has been an integral component of RAS over the years, and the benefits on improved water quality, biosecurity and fish health are well documented [4,5,6,7,8,9]. This strong oxidant improves water quality by reducing organic matter, chemical oxygen demand, dissolved organic carbon, fine particulates, nitrite, water color or even hydrogen sulphide [4,7,10,11]. However, ozone use in aquaculture is mostly used in freshwater systems [12,13]. The use of ozone in brackish and seawater systems presents a challenge as ozone leads to the formation of byproducts (total residual oxidants (TROs)). Ozone dosage and control is still challenging due to limited options in measuring technologies for ozone and TROs [8,14]. Ozone measurements in industrial aquaculture facilities are often not standardized. The most common measurements for ozone level are expressed as either as oxidation-reduction potential (ORP) in millivolts (mV), water transparency/turbidity in varying units or TROs as µg L^−1^ of chlorine (Cl_2_) [8]. Spiliotopoulou [6] presented a new approach by using fluorescence organic matter degradation for monitoring ozone, which is presently not used commercially. Bromide (Br^−^) is a common compound in seawater and has no negative effects on marine organisms [15]. However, upon contact with ozone, bromides form free bromines (HOBr, BrO^−^), where bromine (BrO^−^) is very toxic. Around 50% mortality was recorded within 96 h in rainbow trout (*Oncorhynchus mykiss*) exposed to 68 μg L^−1^ Cl_2_ [16]. Ammonia produced from fish metabolism can react with free bromines, hence inhibiting the formation of high amounts of bromate and bromoform. Artificial seawater can be made without bromide but is, for economical and practical reasons, very seldom used in fish production. Earlier ozone exposure studies on marine organisms in RAS revealed that TRO concentrations higher than 60 μg L^−1^ as Cl_2_ (no ORP provided) in turbot (*Scophthalmus maximus* [12,17]), 30–50 μg L^−1^ as Cl_2_ (~320–350 mV) in European seabass (*Dicentrarchus labrax* [18]), 16–23 μg L^−1^ as Cl_2_ (~330 mV) in Atlantic halibut (*Hippoglossus hippoglossus* [19]) and 14–20 μg L^−1^ as Cl_2_ (~400 mV) in European lobster larvae (*Homarus gammarus* [20]) resulted in impaired growth performance and gill physiology, and, eventually, mortality. The safe limit for Atlantic salmon post-smolts in brackish water still has to be established.

The skin and gills represent the indispensable primary barriers of fish, and their responses to environmental changes have significant implications for overall health [21]. Several studies have already revealed the health-related consequences of higher ozone doses in fish, focusing mainly on alterations in histological structures, behavior and liver enzymes [17,18,22,23]. For example, the thickening of the gill surface is an adaptive response to reduce the exposed branchial area to ozone. Ozone-induced gill damage may also lead to internal hypoxia, hematological adaptations like increase in hematocrit that counteracts less oxygen uptake abilities and problems in ion regulation [7,23,24,25,26]. Our current knowledge on the physiological alterations leading to ozone-related mortality in fish is fragmentary, especially the molecular responses at the mucosa (e.g., oxidative stress) where a direct tissue–ozone interaction occurs [17,27,28,29,30,31].

In this study, we investigated the effects of ozone application in Atlantic salmon reared in brackish water. We employed a flow-through system to isolate more efficiently the impact of different ozone doses. To our knowledge, this is the first report of this kind. The results presented here provide the physiological thresholds for ozone use that will be valuable in optimizing land-based brackish water/seawater water treatment protocols for salmon.

## 2. Results

### 2.1. Mortality

At Day 6, an average cumulative mortality of 36% was recorded in the very high group (Figure 1). The experimental group reached the humane end-point and hence was terminated. The first mortality in the high group was recorded at Day 5 and mortality increased considerably until Day 7. Mortality seemed to plateau from Day 8 until termination, registering end-of-experiment mortality of 33%. Dead fish were first registered at Day 8 in the medium group; nonetheless, no dramatic increase was observed thereafter, and mortality was recorded no greater than 1% at Day 10. No mortality was recorded in both the low and control groups (Figure 1).

### 2.2. Blood

The hematocrit (Hct) level was the highest in the high group compared to other treatments, and it was significantly different from the control, low and very high groups (Table 1). Different ozone doses did not significantly affect the serum sodium ion (Na^+^) concentration. For serum chloride (Cl^−^) ion level, no significant differences were identified amongst the control, low, medium and high groups. The level of serum Cl^−^ was lowest in the very high group and was significantly different compared with the control and medium groups. The magnesium (Mg^2+^) ion concentration was highest in the very high group and the level was significantly higher compared with the other groups, except in the high group.

### 2.3. Histology

Figure 2 shows the prevalence of histopathological changes in the gills following exposure to different ozone doses. All pathologies were identified in all treatment groups, except necrosis which was only observed in the very high group. Cases of hyperplasia, hypertrophy and lifting did not show significant differences amongst treatments. In the low group, around 60% of the evaluated lamellae were not affected, while the remaining 40% displayed the key pathologies, where clubbing was the most prevalent. The percentage of unaffected lamellae was reduced by at least 15% in the medium group compared with the control and low groups. Even though some of the cases such as hypertrophy and hyperplasia increased, the changes were not significantly different from the control and low groups. Lamellar fusion (1.7% of the cases evaluated) was observed in the medium group but not from the control and low groups. Only about 30% of the evaluated lamellae were unaffected in the high group, which means a substantial decrease of at least 37% compared with the 3 lower ozone doses. Cases of clubbing were significantly higher in the high group compared with the control group. Lamellar fusion increased significantly by 6 times in the high group compared with the medium group. For the very high group, around 32% of the evaluated lamellae were unaffected. Cases of clubbing and hyperplasia were also documented, but necrosis was the most pronounced pathological change, accounting to around 10% of the evaluated lamellae. Ozone treatment did not significantly affect the skin epidermal surface morphology, and both general and surface appearance remained unchanged (Appendix A).

### 2.4. Gene Expression Profiles in the Skin and Gills

The transcript level of *mnsod* in the skin was significantly affected by the treatments, where increasing ozone doses resulted in downregulation (Figure 3c). Skin *mnsod* expression was significantly downregulated in the high and very high groups compared with the control. In addition, the transcription in these two high groups was significantly lower compared with the low but not with the medium group. Two antioxidant genes (i.e., *gr* and *cu/znsod*) were significantly affected by ozone treatments and increasing doses resulted in elevated gene expression (Figure 3a,d). Branchial *gr* expression was significantly higher in high and very high groups compared with the control group (Figure 3a). No significant inter-treatment differences were observed amongst the low, medium, high and very high groups. The changes in the gill *cu/znsod* expression followed an almost similar tendency (Figure 3b). The transcript level of *cu/znsod* was significantly elevated by at least 2-fold in the medium, high and very high groups compared with the control.

The expression of genes coding for immunoglobulins and antimicrobial peptides was not significantly affected by the treatments, except with *igt* expression in the gills (Figure 4). Branchial *igt* expression was significantly elevated in the high group compared with the medium group, but the increase was not significantly different with the other groups, including the control (Figure 4a).

The transcription of both cytokine markers (i.e., *il1β*, *il10*) demonstrated an almost identical expression profile in the gills where increasing ozone doses resulted in a significant increase in gene expression (Figure 5). The expression of both cytokines was highest in the very high group; *il1β* expression was at least 4-fold higher whereas *il10* was at least 2-fold higher in this group compared with the other treatment groups.

Ozone treatment significantly affected the expression of *hsp70* in the skin but not in the gills (Figure 6A,a). *Hsp70* transcript level was significantly elevated in medium and very high groups compared with the control and low groups (Figure 6A). From the 3 apoptosis-related genes, only the expression of *bax* was significantly impacted by the treatments in both tissues (Figure 6B,b). Cutaneous expression was highest in the very high group, and the transcript level was significantly higher compared with all the other groups except with the high group (Figure 6B). The expression of *bax* in the medium, high and very high groups was significantly elevated compared with the control but not with the low group (Figure 6b). Varying ozone doses did not significantly alter the expression of *casp3a* and *casp7* in both tissues (Figure 6C,c,D,d).

## 3. Discussion

The benefits of ozone in improving water quality and fish health have been proven over the years. The tolerance thresholds to ozone and its byproducts have likewise been documented in many farmed species, although the majority of what is known is based on freshwater systems. This is the first report to document the impacts of ozone in Atlantic salmon reared in brackish water. The short-term trial revealed that higher ozone doses posed significant health and welfare challenges. Nonetheless, the study identified the potential threshold tolerance value, supported by several physiological indicators for ozone use in brackish water-adapted salmon. In this study, a flow-through experiment was chosen as a model to enable better control of ozonation and effectively isolate the impact of ozone in the fish, which can be challenging in RAS. Nonetheless, the values and their associated effects offer insights for RAS-reared salmon as well.

Ozone doses higher than 350 mV resulted in significant mortality in salmon. The kinetics of mortality was apparently different between the high and the very high groups, the two treatments that resulted in end-point mortality higher than 30%. Ozone at 500 mV induced a drastic effect in salmon, resulting in an abrupt, very high single-day mortality 4 days after the start of ozonation. On the other hand, the high group displayed a progressive mortality trend over 10 days indicating that there was a period when fish mounted an adaptive response; however, extended exposure to ozone resulted in death. The mortality data suggest that Atlantic salmon in brackish water are relatively more sensitive to ozone than other aquaculture species. The highest ORP value tested, which resulted in mass mortality, corresponds to 44 µg L^−1^ as Cl_2_. This was lower than the safe limits for turbot (60 µg L^−1^ as Cl_2_) [12], but within the range for seabass (30–50 µg L^−1^ as Cl_2_) [18]. One of the limitations in ozone research in fish, especially in seawater systems, is that TRO and ORP values are not often provided jointly, making comparisons between studies challenging.

The strong impact of higher ozone can be attributed to respiratory impairment. Damaged morpho-structures of the gills will lead to alterations in oxygen uptake, carbon dioxide excretion and ion regulation [24,25,32], and eventually result in internal hypoxia and hypercapnia [18,33] and suppressed growth performance [34,35]. Both high and very high groups had serious gill damage. The measured average hematocrit values in this experiment are slightly (but throughout the treatments systematically) higher than what is expected for Atlantic salmon (44–49% [36]). However, the increased hematocrit in the high group may be related to physiological adaptations to ozone-related hypoxia, possibly because the experimental time could be considered between acute and chronical exposure. An earlier study suggests that the increase in red blood cells (i.e., hematocrit level) and/or hemoglobin in the red blood cells, as well as a lowered blood pH, is a countermeasure during hypoxic conditions [23]. On the other hand, the impairment of the gills, the main site for ion regulation, will compromise ion homeostasis. Therefore, electrolytes provide insights into the physiological disturbances in fish caused by environmental factors such as ozone. We documented that the levels of Cl^−^ were lowest and Mg^2+^ were highest in the serum of fish from the very high O_3_ treatment (measured on Day 6), indicating that ion homeostasis had been significantly altered. Besides the altered electrolyte balance, the changes in these ions indicate that the highest ozone dose induced stress since they have known functions in stress adaptation. A reduced Cl^−^ concentration was probably linked to respiratory acidosis [25].

The severity of the impact of higher ozone doses in salmon was further revealed by histopathological evaluation using key criteria from earlier toxicological exposure studies [12,17,23,33,37]. The overall histopathological state of low and medium groups was similar with the control group, where the majority (>60%) of the evaluated lamellae were not affected and hence were in good gill health status. Lamellar fusion and clubbing were the most pronounced histopathological changes in the gills in relation to increasing ozone doses. These histological alterations decrease the surface area in the gills [23,38], thereby providing a protective mechanism against the dangers of increasing ozone doses. These histological changes in response to increasing ozone doses were in agreement with previous studies in other marine fish [12,17,23], suggesting a similarity in the protective mechanism of fish against elevated ozone doses. Necrosis was a hallmark pathology in the very high group. It is very likely that this morphological change rendered substantial damage to gill functions, hence the observed mass mortality in this group a few days after the start of the dosing. It was also observed on the day that the very high group was terminated, that the fish were heavily ventilating in front of the inlet pipe with oxygenated water. Reiser and colleagues (2010; 2011) did not identify tissue necrotic cases amongst the treatment groups exposed to a similar ozone doses used in this study. This underscores that the sensitivity to ozone is markedly different between species. We could not ascertain whether or how fast salmon can recover from ozone-related gill damage, but an earlier study in white perch suggested that it could take 4 to 14 days [23].

The histological data indicate that skin was not as vulnerable as the gills to ozone-related tissue damage. All the parameters evaluated including epidermal surface morphology and epidermal and dermal thickness remained similar in all treatment groups. This further suggests that skin health, at least within the context of the evaluated parameters, was not compromised by ozone treatments. Structurally, fish skin is far more complex with several layers of interconnected physical barriers (i.e., scales, epidermis, dermis) than the gills, and this might play a part in the observations here.

The overall expression profile reveals that the key markers for defense and stress were markedly regulated in the gills, supporting the documented extensive branchial histo-structural alterations. The gills cover around 0.1–0.4 m^2^ kg^−1^ body weight, representing the largest organ-specific surface interacting with the aquatic environment. This distinct anatomical feature, including the significant structural differences mentioned, provides support as to why gills were significantly affected by ozone and TROs [39,40].

Antioxidants are molecules that play vital functions in protecting organisms from oxidative stress, a state when the levels of reactive oxygen intermediates exceed the capacity of the physiological system to neutralize or sequester them [41]. Ozone is an oxidant, and elevated levels could trigger oxidative stress [42]. Increased levels of antioxidants in response to ozone have been implicated to be a defense mechanism in several fish species against ozone-induced oxidative stress [27,28,43,44]. Glutathione and superoxide are members of the superclass of antioxidant molecules that play an essential role in the neutralization process of free radicals [45]. The elevated levels of *gr* and *cu/znsod* transcripts particularly in the gills at higher ozone levels (≥350 mV) indicate the occurrence of ozone-induced oxidative stress that resulted in a physiological countermeasure with the activation of key antioxidants. A similar antioxidant response was documented in the liver of ozone-exposed fish, as well as in cell cultures [27,28,43]. This indicates that ozone can target the antioxidant system of fish both through direct (i.e., mucosal) and indirect (i.e., hepatic) interactions. Though we cannot conclude the implication of singular *mnsod* downregulation in the overall oxidative status of the skin at higher ozone doses, it could be probable that the ozone concentration was at that level that inhibited the *mnsod* molecule to participate in antioxidant defense.

In mammalian models, ozone exposure resulted in oxidative damage that triggered inflammatory responses at mucosal surfaces [46,47]. “Oxidation-specific epitopes”, molecular complexes generated by oxidative damage, are detected by pattern recognition receptors (PRRs) of innate immunity that will, in turn, activate a series of responses [48], including the cytokine signaling cascade. The gene expression profile demonstrated a marked molecular inflammatory response in the gills; both pro-inflammatory cytokine *il1β* and anti-inflammatory cytokine *il10* marker genes were significantly upregulated in the very high dose. This pronounced response is likely associated with the substantial necrosis in this treatment group. The high expression of these two genes in this group indicates that these signaling molecules might have initiated an inflammatory response on the site of ozone-induced damage, in which these cytokine molecules have critical involvement in coordinating the immune defense, including recruitment of immune cells to the damaged area. *Il-1β* is one of the canonical molecules involved in orchestrating an inflammation cascade in fish [49]. Traditionally, IL-10 inhibits synthesis of cytokines and demonstrates an immunosuppressive function, though its inflammatory role has been indicated in fish [50]. It is possible that there is a tight regulation between these cytokines. The upregulation of both genes indicates a robust inflammatory action, but it could also be a counter-regulation between the pro- and anti- participants of the inflammatory cascade.

Apoptosis or programmed cell death is a critical component in maintaining cellular homeostasis and growth and plays a significant role in immunity and cytotoxicity [51]. *Bax*, a crucial molecular regulator that turns on the apoptotic cascade [52], was upregulated at higher ozone doses in both mucosal tissues. Oxidative stress has been shown to contribute to apoptosis because of the dramatic decrease in cellular energy supply during this episode [53], and this may partly explain the elevated transcription at higher ozone doses. The similar expression profile between gills and skin indicates that *bax*-mediated response has an essential function in ozone toxicity and could be explored as a potential biomarker for mucosal impacts of ozone treatment.

Heat shock proteins (HSPs) are molecular chaperones that play a role in the response to stressful conditions, including oxidative stress [54] caused by ozone treatment [17]. HSPs participate in the maintenance of protein structure and folding, supporting and repairing damaged cytoskeleton elements, among others [55]. The upregulation of *hsp70* could have provided a cytoprotective function against cutaneous oxidative damages from ozone, a possibility that was also implicated in a previous ozone-related study within the gills [17].

## 4. Material and Methods

### 4.1. Fish Husbandry and Experimental Conditions

All fish handling procedures complied with the Guidelines of the European Union (2010/63/EU), as well as with national legislation. The experiment was approved by the Norwegian Food Safety Authority (FOTS ID 14505). The trial was performed at the Nofima Centre for Recirculation in Aquaculture (NCRA) in Sunndalsøra, Norway. Atlantic salmon eggs (Salmobreed strain) were obtained from AS Bolaks hatchery (Eikelandsosen, Norway). Fish were hatched and smoltified in the Nofima hatchery in Sunndalsøra and smolts were raised under standard conditions [3] until they reached an average weight of around 100 g. The fish were then transferred to 500 L (*n* = 15) tanks in a flow-through setup. The water flow rate in the test tanks was set at 20 L min^−1^ (hydraulic retention time = 25 min.) and measured 3 times a week using a flow meter (Typ SE35, Bürkert GmbH & Co. KG, Ingelfingen, Germany). Each tank was stocked with 65 smolts (99.8 ± 7.5 g), giving an initial density of 13 kg/m^3^. Fish were allowed to acclimatize to the rearing conditions for 19 days before the actual ozone exposure began. An initial welfare score [56] sampling (5 fish per tank) verified that the fish were in similar condition and clinically healthy. During this period, fish were fed daily with a commercial diet (Nutra Olympic 3 mm, Skretting, Averøy, Norway; proximate composition: moisture 8%, crude fat 23%, crude protein 49%, ash 10%) administered through a belt feeder (24 h/day) over satiation (i.e., overfed 120%). Oxygen, salinity and temperature were maintained and monitored daily (OxyGuard International, Birkerød, Denmark): dissolved oxygen = above 93% saturation, pH = 7.68 ± 0.02, temperature = 6.7 ± 0.4 °C, and salinity = 11.6 ± 1.3 ‰. The photoperiod was set to 24 light: 0 dark h daily cycle.

### 4.2. O_3_ Dosing and Treatments

Ozone was generated (OZAT Ozone Generator, CFS-14 2G, Ozonia Degrémont Technologies Ltd., Zurich, Switzerland) and injected into a 1.75 m^−3^ header tank. The ORP value in the header tank was controlled online using 2 independent probes (OxyGuard International A/S ORP probes with Redox Manta controllers, Farum, Denmark) and the ORP data were continuously logged every 5 min. The mean value of two ORP probes was used for ozone dosing because of the known limitations of ORP measurements [14]. Total residual oxygen (TRO) levels in the tank were measured spectrophotometrically (PhotoLab^®^6100 VIS, WTW, Weilheim, Germany) using the colorimetric diethyl-p-phenylenediamine (DPD) method (Spectroquant Chlorine Test, Merck KGaA, Darmstadt, Germany), where concentration was given as chlorine equivalents in µg L^−1^ Cl_2_. Salmon were exposed to 5 different ozone treatments measured as ORP and [TRO]:Control = 250 (228.0 ± 9.6) mV [0 μg L^−1^ as Cl_2_];
Low = 300 (282.3 ± 5.5) mV [~10 μg L^−1^ as Cl_2_];
Medium = 350 (347.7 ± 5.5) mV [10.7 ± 1.2 μg L^−1^ as Cl_2_];
High = 425 (425.3 ± 15.3) mV [16.3 ± 1.5 μg L^−1^ as Cl_2_];
Very high = 500 (488.3 ± 14.4) mV [44.3 ± 4.7 μg L^−1^ as Cl_2_];
in triplicate tanks. Different ozone doses were generated by mixing ozonated (OW = 500 mV) and unozonated (UW = 250 mV) water at different rations ahead of fish tanks. Ozone concentration was increased gradually, and the target doses were achieved 3 days after the dosing initiated. The experiment was terminated after 10 days from the start of ozone exposure except for the very high treatment that was ended after 6 days due to high mortalities (Figure 1). The humane end-point was set when an average treatment cumulative mortality exceeded 35%. When this was reached, fish were humanely euthanized with an overdose of tricaine methanesulfonate (MS-222; 0.5 g/L, Finquel vet., Scan Aqua AS, Årnes) and the treatment group was terminated. To ensure that fish were exposed to the target concentration, a handheld ORP meter (Multi 3620 IDS & SenTix ORP-T 900, WTW, Weilheim, Germany) was used to check the level in the experimental tank.

### 4.3. Blood, Gill and Skin

Sampling was performed at day 10 in all treatment groups (end of ozone treatment), except in the very high treatment where it was conducted at day 6 due to a very high mortality, prompting the termination of the experimental group. Five fish were taken from each tank and were euthanized with an overdose of MS-222. Blood was sampled from the caudal vein using vacutainers for serum (Vacuette tube, 0.5 mL CAT Serum Separator Clot Activator) and plasma (3 mL LH Lithium Heparin Greiner Bio-One GmbH/Preanalytics, Frickenhausen, Germany). The blood collected through plasma vacutainers were used to measure the hematocrit level (3520 Haemofuge Heraeus, Hanau, Germany 12,000 RPM, 3 min). Serum was collected by a 10-min centrifugation at 3700 RPM (Beckman Coulter Avanti J-15R, Krefeld, Germany) and stored at −20 °C until analysis for blood ions (Cl^−^, Na^+^, Mg^2+^ [mmoll^−1^] (Pentra C400, Horiba, Japan).

The second gill arch on the left side of the fish was dissected out and divided into two portions. A small section consisting of around 5–10 gill lamella was suspended in RNAlater^®^ (Sigma-Aldrich Norway AS, Oslo, Norway), kept at 4 °C for 24 h and then transferred to −80 °C until RNA extraction. The remaining gill tissue was placed in formalin (CellStor pots, CellPath, Powys, U.K.) for histological processing.

Skin samples (2.0 cm × 0.5 cm) were cut from the left side of the fish half a centimeter under the dorsal fin. Half of the sample was placed in RNAlater^®^ for gene expression analysis and the other half into formalin for histological evaluation.

### 4.4. Histology

Gill and skin samples were embedded in paraffin using a Leica TP1020 Benchtop Histoprocessor with the following program: 70%, 96% and 3× 100% ethanol, 3 × xylene and 2 × paraffin in a 10-h processing duration. The embedded samples were cut into 5 µm sections using a microtome (Leica RM2255, Wetzlar, Germany) and thereafter stained with hematoxylin-eosin (Thermo Fischer Scientific, Chiago, IL, USA) in an automatic staining machine (Leica autostainer XL). Microscopy of the stained sections was carried out on a Zeiss Axio Observer Z1 (Carl Zeiss, Oberkochen, Germany).

All histological measurements were performed using AxioVision Rel 4.8 Digital Image Processing software (Carl Zeiss). For the gill sections, the evaluation was carried out by randomly selecting five filaments (2 upper half, 2 lower half and 1 middle) and then narrowing them down to a region consisting of 20 lamellae. A total of 100 lamellae were analyzed per fish. Histopathological alterations were identified based on 6 key common cases in ozone-related toxicological experiments, namely lamellar clubbing, lamellar fusion, hypertrophy, hyperplasia, epithelial lifting and necrosis [12]. Lamellae that did not show any pathology were denoted as “unaffected”. For the skin sections, the epidermal surface morphology was semi-quantitatively characterized based on two features, i.e., (a) general appearance of the epidermis and (b) surface quality of the epidermis, using a 3-point scale scoring system as described earlier [41,57,58]; where 0 indicates healthy skin while 3 denotes poor skin health (Appendix A). Both the gills and skin evaluations were performed under blind assessment to ensure objectivity. In addition, morphometry was performed for the dermis and epidermis where 8 representative measurements were taken from each of the 6 randomly selected fields per fish.

### 4.5. RNA Extraction, cDNA Synthesis and RT-qPCR

RNA was isolated from the gill and skin tissue samples using an Agencourt^®^ RNAdvance™ Tissue Total RNA Purification Kit (Beckman Coulter Inc., Brea, CA, USA) following the manufacturer’s protocol. RNA was measured in a NanoDrop 1000 Spectrophotometer (Thermo Fisher Scientific). cDNA was synthesized from 1000 ng total RNA in a 20 μL reaction volume using TaqMan Reverse Transcription Reagents (Applied Biosystems, Foster City, CA, USA) with random hexamers as reaction primers. The thermocycling conditions followed an incubation at 25 °C for 10 min then at 37 °C for 30 min and lastly, at 95 °C for 5 min.

The transcript levels of the target and reference genes (Table 2) were quantified in a QuantStudio5 real-time PCR system (Applied Biosystems) using the PowerUp™ SYBR™ Green master chemistry (Applied Biosystems). The qPCR reaction mixture included 4 μL 1:10 dilution of cDNA, 5 μL SYBR™ Green Master and 1 μL of forward/reverse primer (10 μM). All samples were run in duplicate, including minus reverse transcriptase and no template controls. The thermocycling protocol included a pre-incubation at 95 °C for 2 min, amplification with 40 cycles at 95 °C for 1 s and 60 °C for 30 s, and a dissociation step series of 95 °C for 15 s, 60 °C for 1 min, and 95 °C for 15 s. A five-point standard curve of 2-fold dilution series was prepared from pooled cDNA to calculate the amplification efficiencies. Three reference genes were tested for their suitability for normalization of the expression data: *Elongation factor 1a* (*eef1a*), *acidic ribosomal protein* (*arp*) and *60S ribosomal protein L13* (*rpl13*) [59]. The geNorm software was used to identify the most stable genes in a given tissue. *eef1a* and *rpl13* were the two most stable genes in the skin while *rpl13* and *arp* were found to be stably expressed in the gills. The geometric average of the two stable reference genes was used to normalize the expression of the target genes, as described previously [60].

### 4.6. Statistical Analysis

All data are presented as mean value ± standard deviation (mean ± SD). All statistical analyses were carried out in SigmaStat (SystatSoftware, London, UK). For data sets that complied with normality and equal variance requirements, one-way analysis of variance (ANOVA) was carried out followed by Tukey’s multiple comparison test to identify differences between groups. For non-parametric data sets, Kruskal–Wallis one-way ANOVA on ranks followed by Dunn’s multiple comparison test was used instead. The level of significance was set at *p* < 0.05.

## 5. Conclusions

The current study provides the biological consequences in the use of ozone and indicates that 350 mV is the upper safe limit for ozone in salmon in brackish water. Higher ozone doses, particularly at 425 mV and higher, resulted in significant mortality. This dramatic impact was associated with a substantial ozone-induced gill damage that might have affected respiration and ion regulation. The reduced Cl^−^ concentration and increased hematocrit level lends support to the occurrence of internal hypoxia caused by the gill damage. Both histological and molecular data indicate that the gills were much more sensitive and vulnerable to higher ozone doses than the skin. The gills mounted pronounced antioxidative and inflammatory responses as a protective strategy against elevated ozone levels. The data from this study corroborate several earlier studies (freshwater and seawater) indicating that ≤350 mV is likely the safe ORP threshold value for many farmed fish [18]. It is important to emphasize that the responses and threshold identified here are only applicable for *ca* 100 g salmon in a brackish water flow-through system. Other systems with high organic load and ammonia concentrations (i.e., RAS), as well as the size of fish, may result in a different physiological response. Future studies should be directed at understanding the impact of longer exposure times of the identified threshold by using serum liver enzymes as well as additional blood ions, oxidative stress markers and hematological variables.

## Figures and Tables

**Figure 1 ijms-21-05109-f001:**
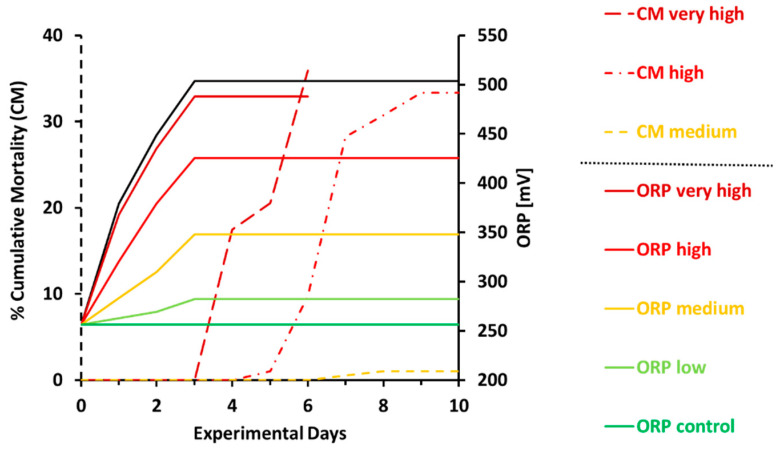
Mean cumulative treatment mortality (CM [%]; *n* = 3; dashed lines) and daily ozone dosage (solid lines) given as ORP = oxidation reduction potential [mV] during the experimental days. Control = 250 (228.0 ± 9.6) mV; low = 300 (282.3 ± 5.5) mV; medium = 350 (347.7 ± 5.5) mV; high = 425 (425.3 ± 15.3) mV; very high = 500 (488.3 ± 14.4) mV. Treatments were adjusted by mixing certain amounts of ozonated water from a header tank with untreated ozone free water. There was a gradual increase over 3 days of ozone dosage in the header tank: day 0 = 256.4 ± 2.3 mV, day 1 = 379.2 ± 22.4 mV, day 2 = 448.7 ± 11.6 mV, day 3 = 503.4 ± 18.5 mV, followed by 8 days of end concentration exposure except for the very high group where the experiment was stopped after 6 days due to high mortalities. Salinity 13‰, temperature 7 °C.

**Figure 2 ijms-21-05109-f002:**
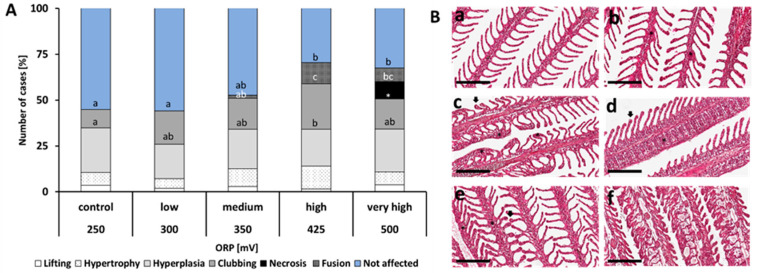
(**A**): Quantification of histopathological changes in the gills of Atlantic salmon exposed to 5 different ozone levels (250, 300, 350, 425 and 500 mV), expressed as oxidation reduction potential (ORP). Key histopathological parameters that were quantified include “not affected”, “fusion”, “necrosis”, “clubbing”, “hyperplasia”, “hypertrophy” and “lifting” (modified from Reiser et al. 2011). Different letters denote statistically significant differences (*p* < 0.5). (**B**): (**a**–**f**) Representative photographs of some of the pathologies in the gills. (**a**) Healthy gill lamellae with well-defined structures; (**b**) hyperplasia (*) at the base of a lamella; (**c**) different degrees of partial fusion (*) of the gill filaments and signs of lamellar clubbing (↓); (**d**) complete fusion of several lamellae (*) and early sign of epithelial lifting(↓); (**e**) signs of hyperplasic and hypertrophy (*) and clubbing and shortening of the lamella (↓); early stage of lamellar fusion from the base can be observed as well; (**f**) loss of brachial structure, heavy degeneration and evidence of necrotic tissues. Scale bar (―) = 200 µm.

**Figure 3 ijms-21-05109-f003:**
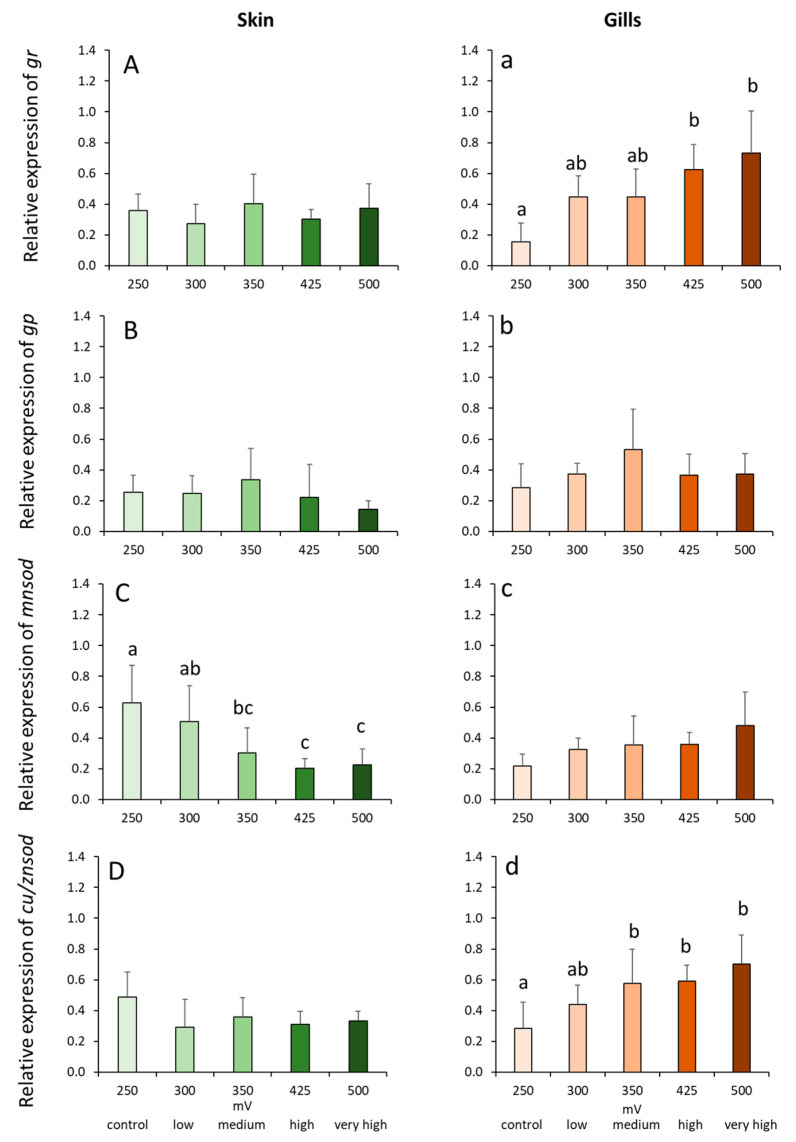
Relative expression of antioxidant defense genes in the skin (**A**–**D**) and gills (**a**–**d**) of Atlantic salmon exposed to varying ORP levels. Values are presented as mean and standard deviation of 6 individual fish per treatment group. Different letters denote significant difference between treatments at *p* < 0.05.

**Figure 4 ijms-21-05109-f004:**
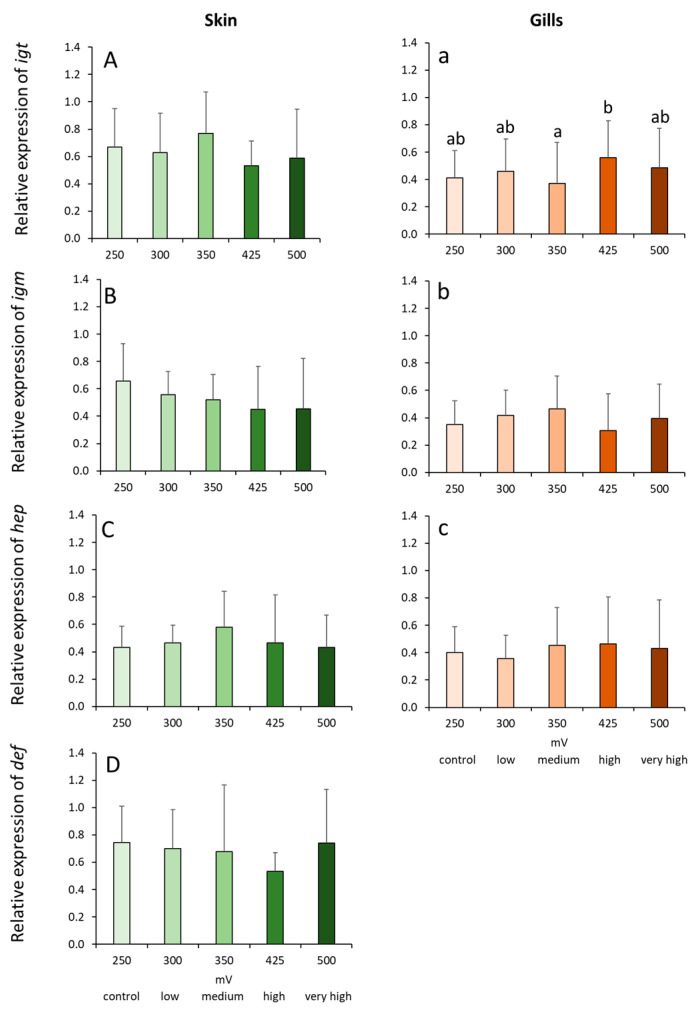
Relative expression of immunoglobulin and antimicrobial peptide genes in the skin (**A**–**D**) and gills (**a**–**c**) of Atlantic salmon exposed to varying ORP levels. Values are presented as mean and standard deviation of 6 individual fish per treatment group. Different letters denote significant differences between treatments at *p* < 0.05. Expression of def in the gills was almost negligible hence was not presented.

**Figure 5 ijms-21-05109-f005:**
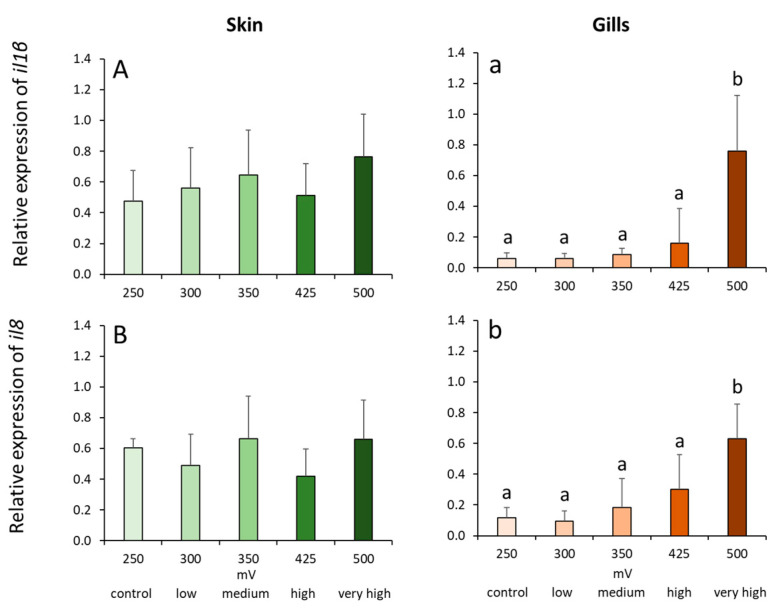
Relative expression of cytokine genes in the skin (**A**,**B**) and gills (**a**,**b**) of Atlantic salmon exposed to varying ORP levels. Values are presented as mean and standard deviation of 6 individual fish per treatment group. Different letters denote significant difference between treatments at *p* < 0.05.

**Figure 6 ijms-21-05109-f006:**
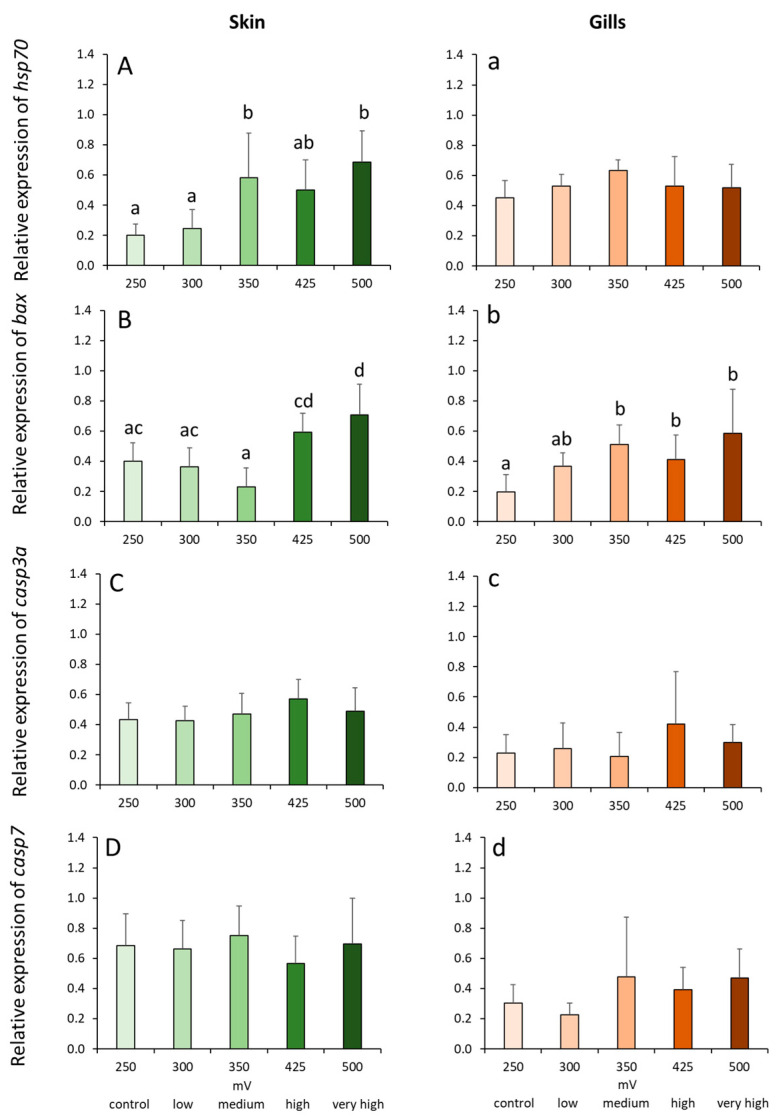
Relative expression of stress and apoptosis-related genes in the skin (**A**–**D**) and gills (**a**–**d**) of Atlantic salmon exposed to varying ORP levels. Values are presented as mean and standard deviation of 6 individual fish per treatment group. Different letters denote significant difference between treatments at *p* < 0.05.

**Table 1 ijms-21-05109-t001:** Mean (±SD) hematocrit (Hct, *n* = 15) (%) and selected blood ion concentrations (Cl, Na, Mg, *n* = 6) (mmol L^−1^) measured in Atlantic salmon post smolts exposed to 5 different ozone treatments (expressed as oxidation reduction potential (ORP) (mV)). * The very high (500 mV) treatment was sampled 5 days earlier.

Treatment	ORP (mV)	Hct	Cl^−^	Na+	Mg^2+^
very high *	500	52.9 ± 5.9 ^a^	121.0 ± 8.1 ^b^	143.5 ± 26.7	1.4 ± 0.2 ^b^
high	425	60.7 ± 5.6 ^b^	127.3 ± 3.3 ^ab^	159.3 ± 2.4	1.2 ± 0.1 ^ab^
medium	350	56.1 ± 6.0 ^ab^	128.2 ± 1.2 ^a^	161.5 ± 2.9	1.1 ± 0.1 ^a^
low	300	51.6 ± 4.7 ^a^	127.0 ± 1.7 ^ab^	162.7 ± 2.3	1.1 ± 0.1 ^a^
control	250	53.7 ± 3.2 ^a^	128.0 ± 1.9 ^a^	161.8 ± 1.2	1.1 ± 0.2 ^a^

Superscript letters indicate significant differences between treatment means (Tukey’s post-hoc test, *p* ˂ 0.05).

**Table 2 ijms-21-05109-t002:** Primers used in the study.

Gene name	Abbreviation	Sequence (5′–3′)	Reference
*glutathione reductase*	*gr*-F	CCAGTGATGGCTTTTTTGAACTT	[61]
	*gr*-R	CCGGCCCCCACTATGAC	
*cu/zn superoxide dismutase*	*cu/znsod*-F	CCACGTCCATGCCTTTGG	
	*cu/znsod*-R	TCAGCTGCTGCAGTCACGTT	
*mn superoxide dismutase*	*mnsod*-F	GTTTCTCTCCAGCCTGCTCTAAG	
	*mnsod*-R	CCGCTCTCCTTGTCGAAGC	
*glutathione peroxidase*	*gp*-F	GATTCGTTCCAAACTTCCTGCTA	
	*gp*-R	GCTCCCAGAACAGCCTGTTG	
*interleukin 1β*	*il1b*-F	AGGACAAGGACCTGCTCAACT	[50]
	*il1b*-R	CCGACTCCAACTCCAACACTA	
*interleukin 10*	*il10*-F	GGGTGTCACGCTATGGACAG	
	*il10*-R	TGTTTCCGATGGAGTCGATG	
*immunoglobulin m*	*igm*-F	TGAGGAGAACTGTGGGCTACACT	[62]
	*igm*-R	TGTTAATGACCACTGAATGTGCAT	
*immunoglobulin t*	*igt*-F	GGTGGTCATGGACGTACTATTT	
	*igt*-R	CCTGTGCAGGCTCATATCTT	
*beta-defensin 1-like*	*defb1l*-F	ATTTAGAAGACGTGGGCG	
	*defb1l*-R	GGATGCTCAAACTACAGTGG	
*heat shock protein 70*	*hsp70*-F	CCCCTGTCCCTGGGTATTG	[61]
	*hsp70*-R	CACCAGGCTGGTTGTCTGAGT	
*bcl-2 associated x protein*	*bax*-F	TGACAGATTTCATCTACGAGCGGG	[63]
	*bax*-R	GCCATCCAGCTCATCTCCAATCT	
*caspase 3a*	*casp3a*-F	ACAGCAAAGAGCTAGAGGTCCAACAC	
	*casp3a*-R	AAAGCCAGGAGAGTTTGACGCAG	
*caspase 7*	*casp7*-F	AGCGAGTGGGCAAGTGCATCA	
	*casp7*-R	CGTCGAAGCCCAGGCTCTTAAA	
*elongation factor alpha-1*	*ef1a*-F	GAATCGGCTATGCCTGGTGAC	[64]
	*ef1a*-R	GGATGATGACCTGAGCGGTG	
*ribosomal protein L13*	*rpl13*-F	CGCTCCAAGCTCATCCTCTTCCC	
	*rpl13*-R	CCATCTTGAGTTCCTCCTCAGTGC	
*acidic ribosomal protein*	*arp*-F	TCATCCAATTGCTGGATGACTATC	[65]
	*arp*-R	CTTCCCACGCAAGGACAGA	

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
