# Peer review of "The Effects of Ozone on Atlantic Salmon Post-Smolt in Brackish Water—Establishing Welfare Indicators and Thresholds"

_ijms, 2020, doi:10.3390/ijms21145109_

Round 1

Reviewer 1 Report

The manuscript of Stiller et al reported the effects of ozone on Atlantic salmon post-smolt in brackish water. There is growing demand for land-based sea/brackish water recirculating salmon farming systems. In line with that this is a timely and well needed research which investigates safe thresholds and welfare indicators in a brackish water flow-through system. The study employed appropriate methodology including physiological, histological and molecular methods as well as relevant statistics to investigate the impact of five different levels of ozone in a brackish water flow-through system. The finding will be valuable in optimizing land-based brackish water/seawater water treatment protocols for salmon.

The manuscript is well written and easy to understand.

I have few minor comments

Abstract: Line 12-13

 At present, it is predominantly used in freshwater systems due to the high risk of toxic residual oxidants in brackish water and seawater.

The sentence is confusing to the readers. Do the authors mean that the high risk of toxic residual oxidants exposure of salmon in brackish water and seawater?

Discussion: Line 421-423

The upregulation of hsp70 could have provided a cytoprotective function against cutaneous oxidative damages from ozone, a possibility that was also implicated in a previous ozone-related study [16].

This sentence is misleading as Reiser, S., et al 2011 did not studied skin hsp70 and observed differences in gill hsp70 expressions in ozone treated group compared to controls on day 1 and 7 while present study did not observe any changes in expression of hsp70 in the gills compared to the control.

Please be specific

Moreover, they observed a decrease in hsp70 response with time, returning to the normal Level by day 21. One of the possible reasons for not observing a difference in expression of hsp70 in gills in the present study could be due to the exhaustion of the hsp70 response similar to the referred study.

Conclusion

The authors did not mention anything about the skin in the conclusion. It is also important to include that the skin was less affected by different levels of ozone treatments

Author Response

Reviewer 1

Line 14

 At present, it is predominantly used in freshwater systems due to the high risk of toxic residual oxidants in brackish water and seawater.

The sentence is confusing to the readers. Do the authors mean that

 the high risk of toxic residual oxidants exposure of salmon in brackish water and seawater?

  • We agree to the reviewers comment. For clarification we added “exposure”

At present, it is predominantly used in freshwater systems due to the high risk of toxic residual oxidants exposure in brackish water and seawater”

Discussion: Line 421-423

The upregulation of hsp70 could have provided a cytoprotective function against cutaneous oxidative damages from ozone, a possibility that was also implicated in a previous ozone-related study [16].

This sentence is misleading as Reiser, S., et al 2011 did not studied skin hsp70 and observed differences in gill hsp70 expressions in ozone treated group compared to controls on day 1 and 7 while present study did not observe any changes in expression of hsp70 in the gills compared to the control.

Please be specific

Moreover, they observed a decrease in hsp70 response with time, returning to the normal Level by day 21. One of the possible reasons for not observing a difference in expression of hsp70 in gills in the present study could be due to the exhaustion of the hsp70 response similar to the referred study.

  • That is true. To clarify this, we added that in the Reiser et al Paper the expression was measured in the Gills not the skin.

Linie 443

Conclusion

The authors did not mention anything about the skin in the conclusion. It is also important to include that the skin was less affected by different levels of ozone treatments

  • We made this clearer by mention it more precisely

  “The reduced Cl- concentration and increased hematocrit level lend support to the occurrence of internal hypoxia caused by the gill damage. Both histological and molecular data indicate that the gills were much more sensitive and vulnerable to higher ozone doses than the skin”.

Reviewer 2 Report

This manuscript entitles “The effects of ozone on Atlantic salmon post-smolt in brackish water establishing welfare indicators and thresholds” evaluated the effects of ozone application in Atlantic salmon reared in brackish water using a flow-through system to isolate more efficiently the impact of different ozone doses.  The use of ozone with re-circulating water in aquaculture systems allows to improved water quality and reduced level of waterborne diseases. Thus, this research represents a valid contribution to the knowledge of fish physiology response in Atlantic salmon in order to prevent certain diseases in farmed species and to guarantee the health and well-being of farmed fish Based on that, I consider the manuscript with interest to be publish. Nevertheless, before it can be published in its final form, I have some concerns that need to be addressed.

The introduction is relevant but it necessary to insert new references and reorganize it.

Material and methods should be improve with more information

The discussion, in the light of the obtained results and of knowledge from is appropriate but it necessary to improve some part. Eliminate the figures and insert the results in tables in order to understand the results obtained.

According to my opinion, the manuscript can be accepting for publication after minor revision.

Below some corrections are reported.

Introduction:

please describe the control and effects of ozone on water quality:

See:

Spiliotopoulou A, Rojas-Tirado P, Chhetri RK, et al. Ozonation control and effects of ozone on water quality in recirculating aquaculture systems. Water Res. 2018;133:289-298. doi:10.1016/j.watres.2018.01.032

The authors state that "many researchs focusing mainly on alterations in histological structures, behavior and liver enzymes ...

Describe the blood parameters that are influenced by ozone treatment. For example, hematological parameters are related to DO so are important for fish health and welfare. See:

Fazio F. 2019. Fish hematology analysis as an important tool of aquaculture: A review. Aquaculture, 500, 237-242.

Oxidative stress parameters also:

Biller JD, Takahashi LS. Oxidative stress and fish immune system: phagocytosis and leukocyte respiratory burst activity. Anais da Academia Brasileira de Ciencias. 2018 Oct-Dec;90(4):3403-3414. DOI: 10.1590/0001-3765201820170730.

Fazio F, Piccione G, Saoca C, Caputo A, Cecchini S (2015) Assessment of oxidative stress in Flathead mullet (Mugil cephalus) and Gilthead sea bream (Sparus aurata). Vet Med 60:2015–2691

.2. Materials and methods

2.1. Fish Husbandry and experimental conditions

All fish were clinically healthy?

During acclimation the photoperiod was set to 24 Light and 0 dark…Why? 

This is an artificial photoperiod that does not correspond to real breeding conditions or natural conditions. We know that the photoperiod is the main synchronizer of all the physiological functions of animals which are regulated by the suprachiasmatic nucleus of the hypothalamus. This photoperiod could mask the real effect of the treatment.

Line 94 for use of commercial diet (Nutra Olympic 3 mm, Skretting, Averøy, Norway) please add in table the composition

How was the health status of the fish assessed?

All samples analyzed in triplicates?

The authors state “ Salmon was exposed o to 5 different ozone treatments …..”  Are there  previous studies that have used this protocol?

Figure3 : improve the quality . the data shown in the histograms for the skin (B and D) and for the gills (b and c) seem to change during treatment. Check the significance obtained or modify the histograms.

Discussion

In discussions the authors state “ The increased haematocrit in the high group may be related to physiological adaptations to ozone related hypoxia.” What is the reference value in salmon for Hct? physiological adaptation in a short period of experiments may not even manifest, in this case it is better to indicate the term of adjustment (acute respoonse). The difference in blood parameters shown in the table could be a normal variation from the physiological range. Indicate the data in the literature. The sampling was always done at the same time ???

The conclusions are too speculative with respect to the results obtained; the modifications induced by ozone treatment concern only one haematological parameter(Hct) and blood ions (Cl- , Na+ and Mg++) . It would be necessary to write that this is a preliminary research and that in the future it is necessary to evaluate other blood parameters (RBC, Hgb, MCV, MCHC and MCH) together Oxidative stress parameters and serum liver enzyme that will complete the experimental investigation

This manuscript entitles “The effects of ozone on Atlantic salmon post-smolt in brackish water establishing welfare indicators and thresholds” evaluated the effects of ozone application in Atlantic salmon reared in brackish water using a flow-through system to isolate more efficiently the impact of different ozone doses.  The use of ozone with re-circulating water in aquaculture systems allows to improved water quality and reduced level of waterborne diseases. Thus, this research represents a valid contribution to the knowledge of fish physiology response in Atlantic salmon in order to prevent certain diseases in farmed species and to guarantee the health and well-being of farmed fish Based on that, I consider the manuscript with interest to be publish. Nevertheless, before it can be published in its final form, I have some concerns that need to be addressed.

The introduction is relevant but it necessary to insert new references and reorganize it.

Material and methods should be improve with more information

The discussion, in the light of the obtained results and of knowledge from is appropriate but it necessary to improve some part. Eliminate the figures and insert the results in tables in order to understand the results obtained.

According to my opinion, the manuscript can be accepting for publication after moderate revision.

Below some corrections are reported.

Introduction:

please describe the control and effects of ozone on water quality:

See:

Spiliotopoulou A, Rojas-Tirado P, Chhetri RK, et al. Ozonation control and effects of ozone on water quality in recirculating aquaculture systems. Water Res. 2018;133:289-298. doi:10.1016/j.watres.2018.01.032

The authors state that "many researchs focusing mainly on alterations in histological structures, behavior and liver enzymes ...

Describe the blood parameters that are influenced by ozone treatment. For example, hematological parameters are related to DO so are important for fish health and welfare. See:

Fazio F. 2019. Fish hematology analysis as an important tool of aquaculture: A review. Aquaculture, 500, 237-242.

Oxidative stress parameters also:

Biller JD, Takahashi LS. Oxidative stress and fish immune system: phagocytosis and leukocyte respiratory burst activity. Anais da Academia Brasileira de Ciencias. 2018 Oct-Dec;90(4):3403-3414. DOI: 10.1590/0001-3765201820170730.

Fazio F, Piccione G, Saoca C, Caputo A, Cecchini S (2015) Assessment of oxidative stress in Flathead mullet (Mugil cephalus) and Gilthead sea bream (Sparus aurata). Vet Med 60:2015–2691

.2. Materials and methods

2.1. Fish Husbandry and experimental conditions

All fish were clinically healthy?

During acclimation the photoperiod was set to 24 Light and 0 dark…Why? 

This is an artificial photoperiod that does not correspond to real breeding conditions or natural conditions. We know that the photoperiod is the main synchronizer of all the physiological functions of animals which are regulated by the suprachiasmatic nucleus of the hypothalamus. This photoperiod could mask the real effect of the treatment.

Line 94 for use of commercial diet (Nutra Olympic 3 mm, Skretting, Averøy, Norway) please add in table the composition

How was the health status of the fish assessed?

All samples analyzed in triplicates?

The authors state “ Salmon was exposed o to 5 different ozone treatments …..”  Are there  previous studies that have used this protocol?

Figure3 : improve the quality . the data shown in the histograms for the skin (B and D) and for the gills (b and c) seem to change during treatment. Check the significance obtained or modify the histograms.

Discussion

In discussions the authors state “ The increased haematocrit in the high group may be related to physiological adaptations to ozone related hypoxia.” What is the reference value in salmon for Hct? physiological adaptation in a short period of experiments may not even manifest, in this case it is better to indicate the term of adjustment (acute respoonse). The difference in blood parameters shown in the table could be a normal variation from the physiological range. Indicate the data in the literature. The sampling was always done at the same time ???

The conclusions are too speculative with respect to the results obtained; the modifications induced by ozone treatment concern only one haematological parameter(Hct) and blood ions (Cl- , Na+ and Mg++) . It would be necessary to write that this is a preliminary research and that in the future it is necessary to evaluate other blood parameters (RBC, Hgb, MCV, MCHC and MCH) together Oxidative stress parameters and serum liver enzyme that will complete the experimental investigation

This manuscript entitles “The effects of ozone on Atlantic salmon post-smolt in brackish water establishing welfare indicators and thresholds” evaluated the effects of ozone application in Atlantic salmon reared in brackish water using a flow-through system to isolate more efficiently the impact of different ozone doses.  The use of ozone with re-circulating water in aquaculture systems allows to improved water quality and reduced level of waterborne diseases. Thus, this research represents a valid contribution to the knowledge of fish physiology response in Atlantic salmon in order to prevent certain diseases in farmed species and to guarantee the health and well-being of farmed fish Based on that, I consider the manuscript with interest to be publish. Nevertheless, before it can be published in its final form, I have some concerns that need to be addressed.

The introduction is relevant but it necessary to insert new references and reorganize it.

Material and methods should be improve with more information

The discussion, in the light of the obtained results and of knowledge from is appropriate but it necessary to improve some part. Eliminate the figures and insert the results in tables in order to understand the results obtained.

According to my opinion, the manuscript can be accepting for publication after moderate revision.

Below some corrections are reported.

Introduction:

please describe the control and effects of ozone on water quality:

See:

Spiliotopoulou A, Rojas-Tirado P, Chhetri RK, et al. Ozonation control and effects of ozone on water quality in recirculating aquaculture systems. Water Res. 2018;133:289-298. doi:10.1016/j.watres.2018.01.032

The authors state that "many researchs focusing mainly on alterations in histological structures, behavior and liver enzymes ...

Describe the blood parameters that are influenced by ozone treatment. For example, hematological parameters are related to DO so are important for fish health and welfare. See:

Fazio F. 2019. Fish hematology analysis as an important tool of aquaculture: A review. Aquaculture, 500, 237-242.

Oxidative stress parameters also:

Biller JD, Takahashi LS. Oxidative stress and fish immune system: phagocytosis and leukocyte respiratory burst activity. Anais da Academia Brasileira de Ciencias. 2018 Oct-Dec;90(4):3403-3414. DOI: 10.1590/0001-3765201820170730.

Fazio F, Piccione G, Saoca C, Caputo A, Cecchini S (2015) Assessment of oxidative stress in Flathead mullet (Mugil cephalus) and Gilthead sea bream (Sparus aurata). Vet Med 60:2015–2691

.2. Materials and methods

2.1. Fish Husbandry and experimental conditions

All fish were clinically healthy?

During acclimation the photoperiod was set to 24 Light and 0 dark…Why? 

This is an artificial photoperiod that does not correspond to real breeding conditions or natural conditions. We know that the photoperiod is the main synchronizer of all the physiological functions of animals which are regulated by the suprachiasmatic nucleus of the hypothalamus. This photoperiod could mask the real effect of the treatment.

Line 94 for use of commercial diet (Nutra Olympic 3 mm, Skretting, Averøy, Norway) please add in table the composition

How was the health status of the fish assessed?

All samples analyzed in triplicates?

The authors state “ Salmon was exposed o to 5 different ozone treatments …..”  Are there  previous studies that have used this protocol?

Figure3 : improve the quality . the data shown in the histograms for the skin (B and D) and for the gills (b and c) seem to change during treatment. Check the significance obtained or modify the histograms.

Discussion

In discussions the authors state “ The increased haematocrit in the high group may be related to physiological adaptations to ozone related hypoxia.” What is the reference value in salmon for Hct? physiological adaptation in a short period of experiments may not even manifest, in this case it is better to indicate the term of adjustment (acute respoonse). The difference in blood parameters shown in the table could be a normal variation from the physiological range. Indicate the data in the literature. The sampling was always done at the same time ???

The conclusions are too speculative with respect to the results obtained; the modifications induced by ozone treatment concern only one haematological parameter(Hct) and blood ions (Cl- , Na+ and Mg++) . It would be necessary to write that this is a preliminary research and that in the future it is necessary to evaluate other blood parameters (RBC, Hgb, MCV, MCHC and MCH) together Oxidative stress parameters and serum liver enzyme that will complete the experimental investigation

This manuscript entitles “The effects of ozone on Atlantic salmon post-smolt in brackish water establishing welfare indicators and thresholds” evaluated the effects of ozone application in Atlantic salmon reared in brackish water using a flow-through system to isolate more efficiently the impact of different ozone doses.  The use of ozone with re-circulating water in aquaculture systems allows to improved water quality and reduced level of waterborne diseases. Thus, this research represents a valid contribution to the knowledge of fish physiology response in Atlantic salmon in order to prevent certain diseases in farmed species and to guarantee the health and well-being of farmed fish Based on that, I consider the manuscript with interest to be publish. Nevertheless, before it can be published in its final form, I have some concerns that need to be addressed.

The introduction is relevant but it necessary to insert new references and reorganize it.

Material and methods should be improve with more information

The discussion, in the light of the obtained results and of knowledge from is appropriate but it necessary to improve some part. Eliminate the figures and insert the results in tables in order to understand the results obtained.

According to my opinion, the manuscript can be accepting for publication after moderate revision.

Below some corrections are reported.

Introduction:

please describe the control and effects of ozone on water quality:

See:

Spiliotopoulou A, Rojas-Tirado P, Chhetri RK, et al. Ozonation control and effects of ozone on water quality in recirculating aquaculture systems. Water Res. 2018;133:289-298. doi:10.1016/j.watres.2018.01.032

The authors state that "many researchs focusing mainly on alterations in histological structures, behavior and liver enzymes ...

Describe the blood parameters that are influenced by ozone treatment. For example, hematological parameters are related to DO so are important for fish health and welfare. See:

Fazio F. 2019. Fish hematology analysis as an important tool of aquaculture: A review. Aquaculture, 500, 237-242.

Oxidative stress parameters also:

Biller JD, Takahashi LS. Oxidative stress and fish immune system: phagocytosis and leukocyte respiratory burst activity. Anais da Academia Brasileira de Ciencias. 2018 Oct-Dec;90(4):3403-3414. DOI: 10.1590/0001-3765201820170730.

Fazio F, Piccione G, Saoca C, Caputo A, Cecchini S (2015) Assessment of oxidative stress in Flathead mullet (Mugil cephalus) and Gilthead sea bream (Sparus aurata). Vet Med 60:2015–2691

.2. Materials and methods

2.1. Fish Husbandry and experimental conditions

All fish were clinically healthy?

During acclimation the photoperiod was set to 24 Light and 0 dark…Why? 

This is an artificial photoperiod that does not correspond to real breeding conditions or natural conditions. We know that the photoperiod is the main synchronizer of all the physiological functions of animals which are regulated by the suprachiasmatic nucleus of the hypothalamus. This photoperiod could mask the real effect of the treatment.

Line 94 for use of commercial diet (Nutra Olympic 3 mm, Skretting, Averøy, Norway) please add in table the composition

How was the health status of the fish assessed?

All samples analyzed in triplicates?

The authors state “ Salmon was exposed o to 5 different ozone treatments …..”  Are there  previous studies that have used this protocol?

Figure3 : improve the quality . the data shown in the histograms for the skin (B and D) and for the gills (b and c) seem to change during treatment. Check the significance obtained or modify the histograms.

Discussion

In discussions the authors state “ The increased haematocrit in the high group may be related to physiological adaptations to ozone related hypoxia.” What is the reference value in salmon for Hct? physiological adaptation in a short period of experiments may not even manifest, in this case it is better to indicate the term of adjustment (acute respoonse). The difference in blood parameters shown in the table could be a normal variation from the physiological range. Indicate the data in the literature. The sampling was always done at the same time ???

The conclusions are too speculative with respect to the results obtained; the modifications induced by ozone treatment concern only one haematological parameter(Hct) and blood ions (Cl- , Na+ and Mg++) . It would be necessary to write that this is a preliminary research and that in the future it is necessary to evaluate other blood parameters (RBC, Hgb, MCV, MCHC and MCH) together Oxidative stress parameters and serum liver enzyme that will complete the experimental investigation

Author Response

Line 52-58

please describe the control and effects of ozone on water quality:

See:

Spiliotopoulou A, Rojas-Tirado P, Chhetri RK, et al. Ozonation control and effects of ozone on water quality in recirculating aquaculture systems. Water Res. 2018;133:289-298. doi:10.1016/j.watres.2018.01.032

  • The main effects on water quality where presented in the introduction.
  • We added the main problems of control ozone in aquaculture.

“Ozone dosage and control is still challenging due to limited options in measuring technologies for ozone and TROs [8, 14]. Ozone measurements in industrial aquaculture facilities are often not standardised. The most common measurements for Ozone level are expressed as either as oxidation-reduction potential (ORP) in millivolts (mV), water transparency/turbidity in varying units, or TROs as µg L-1 of chlorine (Cl2) [8]. Spiliotopoulou [6] presented a quite new approach by using fluorescence organic matter degradation for monitoring ozone which is presently not commercially used.”

-----------------------------------------------------------------------------------------

Line 77 - 79

The authors state that "many researchs focusing mainly on alterations in histological structures, behavior and liver enzymes ...

Describe the blood parameters that are influenced by ozone treatment.

For example, hematological parameters are related to DO so are important for fish health and welfare. See:

Fazio F. 2019. Fish hematology analysis as an important tool of aquaculture: A review. Aquaculture, 500, 237-242.

  • We added the reference and give the most important example hematocrit for this publication.

Oxidative stress parameters also:

Biller JD, Takahashi LS. Oxidative stress and fish immune system: phagocytosis and leukocyte respiratory burst activity. Anais da Academia Brasileira de Ciencias. 2018 Oct-Dec;90(4):3403-3414. DOI: 10.1590/0001-3765201820170730.

Fazio F, Piccione G, Saoca C, Caputo A, Cecchini S (2015) Assessment of oxidative stress in Flathead mullet (Mugil cephalus) and Gilthead sea bream (Sparus aurata). Vet Med 60:2015–2691

  • We agree that is useful to mention at least the most important marker (oxidative stress) and cite booth papers. To hold the introduction short we think it would be enough. At the end these papers are not that specific for going more into detail for an ozone experiment.

“Ozone-induced gill damage may also lead to internal hypoxia, hematological adaptations like increase in hematocrit that counteracts less oxygen uptake abilities and problems in ion regulation [7, 23-26]. Our current knowledge on the physiological alterations leading to ozone-related mortality in fish is fragmentary, especially the molecular responses at the mucosa (e.g. oxidative stress) where a direct tissue-ozone interaction occurs [17, 27-31].”

-----------------------------------------------------------------------------------------

Linie 100 – 102

All fish were clinically healthy?

  • Yes. See comment answer below.

How was the health status of the fish assessed?

  • We evaluated the fish intensively (wellfare score) after the adaption phase.
  • Noble, C., Gismervik, K., Iversen, M. H., Kolarevic, J., Nilsson, J., Stien, L. H., & Turnbull, J. F. (2018). Welfare Indicators for farmed Atlantic salmon: tools for assessing fish welfare.

“An initial welfare score [32] sampling (5 fish per tank) verified that the fish where in similar condition and clinically healthy. “

-----------------------------------------------------------------------------------------

Linie 102

Line 94 for use of commercial diet (Nutra Olympic 3 mm, Skretting, Averøy, Norway) please add in table the composition

  • We added the proximate composition. We believe there is no need more infromation in a table that represents a standard salmon feed.

"(Nutra Olympic 3 mm, Skretting, Averøy, Norway ; proximate composition: Moisture 8%, Crude fat 23%, Crude Protein 49%, Ash 10%)"

----------------------------------------------------------------------------------------

Linie 106 - 107

During acclimation the photoperiod was set to 24 Light and 0 dark…Why?

This is an artificial photoperiod that does not correspond to real breeding conditions or natural conditions. We know that the photoperiod is the main synchronizer of all the physiological functions of animals which are regulated by the suprachiasmatic nucleus of the hypothalamus. This photoperiod could mask the real effect of the treatment.

  • The experiment represents fish production conditions.
  • We agree that that the light conditions are not natural. However, It is not uncommon in industrial practice to raise post smolt under 24 hours light conditions.

-----------------------------------------------------------------------------------------

Linie 119 - 123

The authors state “ Salmon was exposed o to 5 different ozone treatments …..” Are there previous studies that have used this protocol?

  • Not exactly these values. However, in the Introduction we mention Studies that been similar to the used values.

-----------------------------------------------------------------------------------------

Linie 124

All samples analyzed in triplicates?

  • Yes. We would guess it should be sufficient if its stated as it is as “in triblicate tanks”. If we refer later that from every tank samples where taken it will imply that all been done in triplicate.

-----------------------------------------------------------------------------------------

Linie 137

All samples analyzed in triplicates?

  • Yes, it is written in section 2.2.

-----------------------------------------------------------------------------------------

Linie 281

Figure 3: improve the quality. the data shown in the histograms for the skin (B and D) and for the gills (b and c) seem to change during treatment. Check the significance obtained or modify the histograms.

  • It is quite common to present gene expression data as a histogram. See Reiser et al.
  • “seem to change during treatment.” Yes, indeed there are tendencies. However, the standard deviations are quite high. The used statistics are quite simple. There are no changes in significance by redoing the statistics.
  • We do not think that there is a benefit for the reader to modify the histograms.

-----------------------------------------------------------------------------------------

Eliminate the figures and insert the results in tables in order to understand the results obtained.

  • We started writing the MS by using tables. We do not agree that this is a benefit for the reader to present the data in a table.

-----------------------------------------------------------------------------------------

Linie 345

In discussions the authors state “ The increased haematocrit in the high group may be related to physiological adaptations to ozone related hypoxia.” What is the reference value in salmon for Hct?

physiological adaptation in a short period of experiments may not even manifest, in this case it is better to indicate the term of adjustment (acute respoonse). The difference in blood parameters shown in the table could be a normal variation from the physiological range. Indicate the data in the literature. The sampling was always done at the same time ???

  • We took at least 3 haematocrit samples from one fish. The variation within the fish is high. Linking the outcome to the statistics it is very likely that what we presentened is a treatment effect and not the normal variation within the group.
  • The sampling was done tank by tank throughout a day. So that every treatment got the same time dependent variation thought a day.

“The measured average hematocrit values in this experiment are slightly but throughout the treatments systematically higher of what is expected for Atlantic salmon (44–49%[48]). However, the increased haematocrit in the high group may be related to physiological adaptations to ozone-related hypoxia possibly because the experimental time could be considered between acute and chronical exposure.”

-----------------------------------------------------------------------------------------

Conclusion

Linie 450

The conclusions are too speculative with respect to the results obtained; the modifications induced by ozone treatment concern only one haematological parameter(Hct) and blood ions (Cl- , Na+ and Mg++) . It would be necessary to write that this is a preliminary research and that in the future it is necessary to evaluate other blood parameters (RBC, Hgb, MCV, MCHC and MCH) together Oxidative stress parameters and serum liver enzyme that will complete the experimental investigation

  • We added the information the last sentence.

“Future studies should be directed at understanding the impact at longer exposure times of the identified threshold by using serum liver enzymes as well as additional blood ions, oxidative stress markers and hematological variables.”

-----------------------------------------------------------------------------------------

Line 52-58

Reviewer 1

please describe the control and effects of ozone on water quality:

See:

Spiliotopoulou A, Rojas-Tirado P, Chhetri RK, et al. Ozonation control and effects of ozone on water quality in recirculating aquaculture systems. Water Res. 2018;133:289-298. doi:10.1016/j.watres.2018.01.032

  • The main effects on water quality where presented in the introduction.
  • We added the main problems of control ozone in aquaculture.

“Ozone dosage and control is still challenging due to limited options in measuring technologies for ozone and TROs [8, 14]. Ozone measurements in industrial aquaculture facilities are often not standardised. The most common measurements for Ozone level are expressed as either as oxidation-reduction potential (ORP) in millivolts (mV), water transparency/turbidity in varying units, or TROs as µg L-1 of chlorine (Cl2) [8]. Spiliotopoulou [6] presented a quite new approach by using fluorescence organic matter degradation for monitoring ozone which is presently not commercially used.”

-----------------------------------------------------------------------------------------

Line 77 - 79

Reviewer 1

The authors state that "many researchs focusing mainly on alterations in histological structures, behavior and liver enzymes ...

Describe the blood parameters that are influenced by ozone treatment.

For example, hematological parameters are related to DO so are important for fish health and welfare. See:

Fazio F. 2019. Fish hematology analysis as an important tool of aquaculture: A review. Aquaculture, 500, 237-242.

  • We added the reference and give the most important example hematocrit for this publication.

Oxidative stress parameters also:

Biller JD, Takahashi LS. Oxidative stress and fish immune system: phagocytosis and leukocyte respiratory burst activity. Anais da Academia Brasileira de Ciencias. 2018 Oct-Dec;90(4):3403-3414. DOI: 10.1590/0001-3765201820170730.

Fazio F, Piccione G, Saoca C, Caputo A, Cecchini S (2015) Assessment of oxidative stress in Flathead mullet (Mugil cephalus) and Gilthead sea bream (Sparus aurata). Vet Med 60:2015–2691

  • We agree that is useful to mention at least the most important marker (oxidative stress) and cite booth papers. To hold the introduction short we think it would be enough. At the end these papers are not that specific for going more into detail for an ozone experiment.

“Ozone-induced gill damage may also lead to internal hypoxia, hematological adaptations like increase in hematocrit that counteracts less oxygen uptake abilities and problems in ion regulation [7, 23-26]. Our current knowledge on the physiological alterations leading to ozone-related mortality in fish is fragmentary, especially the molecular responses at the mucosa (e.g. oxidative stress) where a direct tissue-ozone interaction occurs [17, 27-31].”

-----------------------------------------------------------------------------------------

Linie 100 – 102

Reviewer 1

All fish were clinically healthy?

  • Yes. See comment answer below.

How was the health status of the fish assessed?

  • We evaluated the fish intensively (wellfare score) after the adaption phase.
  • Noble, C., Gismervik, K., Iversen, M. H., Kolarevic, J., Nilsson, J., Stien, L. H., & Turnbull, J. F. (2018). Welfare Indicators for farmed Atlantic salmon: tools for assessing fish welfare.

“An initial welfare score [32] sampling (5 fish per tank) verified that the fish where in similar condition and clinically healthy. “

-----------------------------------------------------------------------------------------

Linie 102

Reviewer 1

Line 94 for use of commercial diet (Nutra Olympic 3 mm, Skretting, Averøy, Norway) please add in table the composition

  • We added the proximate composition. We believe there is no need more infromation in a table that represents a standard salmon feed.

"(Nutra Olympic 3 mm, Skretting, Averøy, Norway ; proximate composition: Moisture 8%, Crude fat 23%, Crude Protein 49%, Ash 10%)"

----------------------------------------------------------------------------------------

Linie 106 - 107

Reviewer 1

During acclimation the photoperiod was set to 24 Light and 0 dark…Why?

This is an artificial photoperiod that does not correspond to real breeding conditions or natural conditions. We know that the photoperiod is the main synchronizer of all the physiological functions of animals which are regulated by the suprachiasmatic nucleus of the hypothalamus. This photoperiod could mask the real effect of the treatment.

  • The experiment represents fish production conditions.
  • We agree that that the light conditions are not natural. However, It is not uncommon in industrial practice to raise post smolt under 24 hours light conditions.

-----------------------------------------------------------------------------------------

Linie 119 - 123

Reviewer 1

The authors state “ Salmon was exposed o to 5 different ozone treatments …..” Are there previous studies that have used this protocol?

  • Not exactly these values. However, in the Introduction we mention Studies that been similar to the used values.

-----------------------------------------------------------------------------------------

Linie 124

Reviewer 1

All samples analyzed in triplicates?

  • Yes. We would guess it should be sufficient if its stated as it is as “in triblicate tanks”. If we refer later that from every tank samples where taken it will imply that all been done in triplicate.

-----------------------------------------------------------------------------------------

Linie 137

Reviewer 1

All samples analyzed in triplicates?

  • Yes, it is written in section 2.2.

-----------------------------------------------------------------------------------------

Linie 281

Reviewer 1:

Figure 3: improve the quality. the data shown in the histograms for the skin (B and D) and for the gills (b and c) seem to change during treatment. Check the significance obtained or modify the histograms.

  • It is quite common to present gene expression data as a histogram. See Reiser et al.
  • “seem to change during treatment.” Yes, indeed there are tendencies. However, the standard deviations are quite high. The used statistics are quite simple. There are no changes in significance by redoing the statistics.
  • We do not think that there is a benefit for the reader to modify the histograms.

-----------------------------------------------------------------------------------------

Reviewer 1

Eliminate the figures and insert the results in tables in order to understand the results obtained.

  • We started writing the MS by using tables. We do not agree that this is a benefit for the reader to present the data in a table.

-----------------------------------------------------------------------------------------

Linie 345

Reviewer 1

In discussions the authors state “ The increased haematocrit in the high group may be related to physiological adaptations to ozone related hypoxia.” What is the reference value in salmon for Hct?

physiological adaptation in a short period of experiments may not even manifest, in this case it is better to indicate the term of adjustment (acute respoonse). The difference in blood parameters shown in the table could be a normal variation from the physiological range. Indicate the data in the literature. The sampling was always done at the same time ???

  • We took at least 3 haematocrit samples from one fish. The variation within the fish is high. Linking the outcome to the statistics it is very likely that what we presentened is a treatment effect and not the normal variation within the group.
  • The sampling was done tank by tank throughout a day. So that every treatment got the same time dependent variation thought a day.

“The measured average hematocrit values in this experiment are slightly but throughout the treatments systematically higher of what is expected for Atlantic salmon (44–49%[48]). However, the increased haematocrit in the high group may be related to physiological adaptations to ozone-related hypoxia possibly because the experimental time could be considered between acute and chronical exposure.”

-----------------------------------------------------------------------------------------

Conclusion

Linie 450

Reviewer 1

The conclusions are too speculative with respect to the results obtained; the modifications induced by ozone treatment concern only one haematological parameter(Hct) and blood ions (Cl- , Na+ and Mg++) . It would be necessary to write that this is a preliminary research and that in the future it is necessary to evaluate other blood parameters (RBC, Hgb, MCV, MCHC and MCH) together Oxidative stress parameters and serum liver enzyme that will complete the experimental investigation

  • We added the information the last sentence.

“Future studies should be directed at understanding the impact at longer exposure times of the identified threshold by using serum liver enzymes as well as additional blood ions, oxidative stress markers and hematological variables.”

-----------------------------------------------------------------------------------------

We tank the reviewer for there time and hope that we address the comments appropriate.